# Protocols for Monitoring Harmful Algal Blooms for Sustainable Aquaculture and Coastal Fisheries in Chile

**DOI:** 10.3390/ijerph17207642

**Published:** 2020-10-20

**Authors:** Kyoko Yarimizu, So Fujiyoshi, Mikihiko Kawai, Luis Norambuena-Subiabre, Emma-Karin Cascales, Joaquin-Ignacio Rilling, Jonnathan Vilugrón, Henry Cameron, Karen Vergara, Jesus Morón-López, Jacquelinne J. Acuña, Gonzalo Gajardo, Oscar Espinoza-González, Leonardo Guzmán, Milko A. Jorquera, Satoshi Nagai, Gemita Pizarro, Carlos Riquelme, Shoko Ueki, Fumito Maruyama

**Affiliations:** 1Office of Industry-Academia-Government and Community Collaboration, Hiroshima University, 1-3-2 Kagamiyama, Higashi-Hiroshima City, Hiroshima 739-8511, Japan; fujiyoshi.so.62w@kyoto-u.jp; 2Graduate School of Human and Environmental Studies, Kyoto University, Yoshidanihonmatsu-cho, Kyoto 606-8501, Japan; kawai.mikihiko.8c@kyoto-u.ac.jp; 3Centro de Estudios de Algas Nocivas (CREAN), Instituto de Fomento Pesquero (IFOP), Padre Harter 547, Puerto Montt 5480000, Chile; luis.norambuena@ifop.cl (L.N.-S.); emma.cascales@ifop.cl (E.-K.C.); vilugront@gmail.com (J.V.); oscar.espinoza@ifop.cl (O.E.-G.); leonardo.guzman@ifop.cl (L.G.); 4Scientific and Biotechnological Bioresource Nucleus (BIOREN-UFRO), Universidad de La Frontera, Ave. Francisco Salazar 01145, Temuco 4811230, Chile; ignacio.rilling@gmail.com (J.-I.R.); jacquelinne.acuna@ufrontera.cl (J.J.A.); milko.jorquera@ufrontera.cl (M.A.J.); 5Centro de Bioinnovacion, Facultad de Ciencias del Mar y Recursos Biologicos, Universidad de Antofagasta, Av. Angamos 601, Antofagasta 1270300, Chile; bastiancameron@gmail.com (H.C.); ceriquelme@gmail.com (C.R.); 6Laboratorio de Genética, Acuicultura & Biodiversidad, Departamento de Ciencias Biológicas y Biodiversidad, Universidad de Los Lagos, Osorno 5290000, Chile; karen.vergara@ulagos.cl (K.V.); ggajardo@ulagos.cl (G.G.); 7Institute of Plant Science and Resources, Okayama University, 2-20-1 Chuo, Kurashiki, Okayama 710-0046, Japan; jesmorlop@gmail.com (J.M.-L.); shokoueki@okayama-u.ac.jp (S.U.); 8Japan Fisheries Research and Education Agency, Fisheries Resources Institute, Fisheries Stock Assessment Center, 2-12-4 Fukuura, Kanazawa-ku, Yokohama, Kanagawa 236-8648, Japan; snagai@affrc.go.jp; 9Laboratorio de toxinas y fitoplancton, IFOP, Enrique Abello 0552, Punta Arenas 6200000, Chile; gemita.pizarro@ifop.cl

**Keywords:** HAB monitoring, harmful algae, phytoplankton, coastal monitoring, metabarcoding analysis, DNA extraction, microscope, pigment assay, toxin assay, nutrient assay

## Abstract

Harmful algae blooms (HABs) cause acute effects on marine ecosystems due to their production of endogenous toxins or their enormous biomass, leading to significant impacts on local economies and public health. Although HAB monitoring has been intensively performed at spatiotemporal scales in coastal areas of the world over the last decades, procedures have not yet been standardized. HAB monitoring procedures are complicated and consist of many methodologies, including physical, chemical, and biological water sample measurements. Each monitoring program currently uses different combinations of methodologies depending on site specific purposes, and many prior programs refer to the procedures in quotations. HAB monitoring programs in Chile have adopted the traditional microscopic and toxin analyses but not molecular biology and bacterial assemblage approaches. Here we select and optimize the HAB monitoring methodologies suitable for Chilean geography, emphasizing on metabarcoding analyses accompanied by the classical tools with considerations including cost, materials and instrument availability, and easiness and efficiency of performance. We present results from a pilot study using the standardized stepwise protocols, demonstrating feasibility and plausibility for sampling and analysis for the HAB monitoring. Such specific instructions in the standardized protocol are critical obtaining quality data under various research environments involving multiple stations, different analysts, various time-points, and long HAB monitoring duration.

## 1. Introduction

Over the years, almost every coast in the world has experienced a number of harmful algae blooms (HABs) [1,2,3]. Some algal species secrete endogenous toxins that lead to damages to local marine ecosystems, socio-economic, and public health, and even non-toxic species can be noxious, causing oxygen depletion by organic matter decomposition from a high biomass accumulation [4,5]. Although mechanisms of HAB are not fully elucidated, the frequency of HAB and their intensity of consequences are expected to increase in the future as global ocean temperatures rise [6,7]. Chile, a country in the southern Pacific Ocean, has been severely affected by HABs damaging the well-known aquaculture and fishery industries with broad socio-ecological impacts [8]. Blooms of *Pseudochattonella verruculosa* and *Alexandrium catenella* during the austral summer of 2016 were particularly severe, causing the largest fish and bivalve mortality rates ever recorded in the world. This resulted in an exportation income loss of USD$800 million and stimulated social strikes on Chiloé island in southern Chile [9,10,11]. Thus, HAB scientists locally and globally are continuously exerting every effort to establish a better and faster coastal monitoring system to predict upcoming HABs that can be used in future strategic remediation.

The current major HAB monitoring strategies rely on microscopic observation of HAB species and toxin analysis. Chile has used these standard practices for decades, and today there are four ongoing monitoring programs of different pursuits: The National Intoxication Prevention and Control Red Tide Program (PNMR) under the Ministry of Health surveils the effect of marine bio-toxins on epidemiology, aiming to avert human illness from the consumption of HAB-derived contaminated marine sources. The Program of the Fishery and Aquaculture Undersecretary by the Ministry of Economy pursues protection of public health, fisheries, aquaculture, and tourism using two studies, monitoring HABs in the Chilean fjords and channels and that of toxins in the Pacific Ocean of south-central Chile. The Bivalves-Mollusks Health Program (PSMB), financed by the mussel farming sectors, ensures the safety of products destined for export markets, and the National Fisheries and Aquaculture Service (SERNAPESCA) is responsible for certificating the products. The Phytoplankton Monitoring Program funded by the private sectors aims to protect salmon farms by providing timely information to minimize the effects of HABs on caged fish. Further, Instituto de Fomento Pesquero (Institute of Fisheries Development, IFOP) routinely monitors 307 stations along fjords and open ocean to provide environmental information associated with phytoplankton assemblages, including cyst abundance in sediments. These traditional approaches help us understand the relationship between local HAB dynamics and environmental factors. However, no HAB programs in Chile have incorporated molecular biology and bacterial assemblage approaches into their monitoring.

Recently, bacteria have been attracting attention as another possible factor that may be involved in HABs. Since Bell and Mitchell reported that specific bacterial communities altered the zone around microalgae in the so-called phycosphere [12,13], an increasing number of studies address this specific and relevant interaction [14,15,16,17,18]. Such ideas propose an attractive hypotheses that mutualistic algal-bacterial association is based on nutrient exchange and has synergetic or antagonistic relationships accounting for bloom development and regression [19]. Thus, our study focuses on gaining more comprehensive HAB knowledge of HAB species with bacteria referred to as a holobiont. To this end, we adopt metabarcoding analysis to understand the microbe assemblages and their importance in the marine environment in Chile. The metabarcoding analysis is a powerful tool for its ability to detect massive taxonomic information in a sample, even with a low abundance of target organisms [20,21,22].

HAB monitoring has been performed in many coastal regions globally; however, the methodologies used for each monitoring program are not standardized. Of all the methodologies, we selected and optimized the protocols for our prospective HAB monitoring project to monitor the water samples on a regular basis at 14 stations in Chile for four years to understand the HAB dynamics with holobiome factors. We describe them stepwise, emphasizing metabarcoding analytics followed by the classical tools of microscopic species observation, meteorological variables, and physical and chemical water properties. To the best of our knowledge, this is the first HAB monitoring protocol described in detailed steps. Such specific instructions in this standardized protocol, even if it seems trivial, are critical components to collect reliable data under various research environments such as multiple stations, different analysts, various time-points, and long study durations. A pilot study was performed on one of the 14 target stations by following these protocols, and the results are discussed herein with respect to feasibility, productivity, and plausibility of the protocols in addition to the justification for selecting specific methodologies. The information obtained using these protocols through the 4-year project is planned to build the HAB forecast system for Chilean waters. We hope that the information provided here lays the groundwork to replicate this holistic monitoring program in other HAB-affected areas worldwide.

## 2. Materials and Methods

Information on materials, reagents and instruments is listed in Appendix Table A1. Always use disposable nitrile gloves without directly touching samples and devices.

### 2.1. Sampling

This section describes a sampling method and sample storage.

#### 2.1.1. Sample Collection

Prepare materials and devices as in Figure 1. Go to sampling points using the GPS coordinator for precise identification of sampling points. Figure 2 shows the 14 stations to be monitored for our prospective project. In the present report, a pilot study was performed at station 3 (Metri) to justify our protocols.Deploy a CTD (an oceanography instrument to measure conductivity, temperature, and pressure of seawater as a function of depth) to a 3-m depth and stabilize it for 5 min. Pull the CTD to the surface quickly and deploy it back to each station’s maximum depth. Pull the CTD from the depth to surface to collect the value of temperature, salinity, dissolved oxygen (DO), and chlorophyll-a (chl-*a*) as a depth function. Report average of values collected from 0 to 10-m depth.Deploy a Secchi disc slowly by 1-m increment until the disc ceases to be visible from the surface. Record the depth of transparency of water.

#### 2.1.2. Water Sample for Assays

Rinse a plastic container three times with seawater at the sampling point. Deploy a 10-cm diameter siphoning hose with both top and bottom seal opened to a 10-m depth. When the hose bottom reaches a 10-m depth, lock the top of the hose. Pull the hose to the surface. Open the top seal and immediately draw off the water sample into the rinsed plastic container to collect at least 5 liters of the sample (go to Section 2.2).

#### 2.1.3. Water Sample for Nutrient Analysis

Separately, draw off approximately 1 L of water sample (to collect well-mixed water from the water column) for dissolved nutrient assay into a 1-L plastic container pre-rinsed with the water sample three times. Filter the water sample on-site through 0.22-μm pore size membrane immediately after sampling. Rinse five sterile 2-mL tubes and fill the tubes with the filtrate. Each tube will be used for the analysis of nitrite, nitrate, phosphate, and silicate. Place the tubes in a labeled plastic bag and store at −20 °C (if freezer is not available on-site, store samples in an icebox to carry back to the lab) until analysis (go to Section 2.3.3 for analysis). Care must be taken to avoid contamination of samples: Avoid forming bubbles in samples to prevent oxidation. Do not fill tubes to the top as water samples expand upon freezing. Transfer samples to the freezer in the shortest possible time. Label clearly to avoid confusion. Filter samples slowly as vigorous filtration may rupture the filter membrane.

### 2.2. Sample Treatment

This section covers methods to treat water samples for phytoplankton, pigment, and metabarcoding analyses. Treat the water sample collected in Section 2.1 on the sampling ship or nearest accessible laboratory within 12 h of sampling.

#### 2.2.1. Phytoplankton Identification and Quantification

Prepare two phytoplankton samples per station, raw and fixed. For raw samples, mix water sample from Section 2.1.2 thoroughly and transfer small volume to a 15-mL plastic tube. Next, transfer 1-mL of the sample by a pipette onto a 1-mL grid-slide (Sedgewick Rafter counting chamber). Record names and quantity of phytoplankton species observed under a microscope using our phytoplankton naming dictionary (Appendix A). The fixed sample is used by the following procedures to identify any missing species that the direct observation could not detect:
Filter 200 mL water sample through 1 μm membrane slowly. Do not apply a vacuum too high. Stop the vacuum, while the filtration bottle top still contains 8–12 mL of the water sample. Collect the concentrated water sample on the filtration bottle top using a pipette. Transfer 8–12 mL concentrated sample to a 15-mL plastic centrifuge tube.Add 100 μL of Lugol to fix the sample. Keep at 5 °C for storage. Identify and count phytoplankton species in a 1-mL grid-slide glass (Sedgewick Rafter counting chamber) under a microscope per our phytoplankton naming dictionary (Appendix A).

#### 2.2.2. Sample Treatment for Pigment Analysis

Filter 1 L of a water sample from Section 1 through GF/F filter. Fold the filtered GF/F paper in half and wrap in an aluminum foil. Store at −20 °C until analysis (go to Section 2.3.2 for analysis).

#### 2.2.3. Sample Workups for Metabarcoding Analysis

Filter 1 liters of a water sample from Section 2.1.2 for 16S-rRNA using a tandem filtration (1 μm pore-size connected to 0.2 μm pore-size Sterivex membrane) to separate free-living and attached bacteria (Figure 3). Filter another 1 liters for 18S-rRNA gene analysis using a single filtration with a 0.2 μm Sterivex membrane (Figure 3). Thus, three filter membranes are the product from one sampling point (Figure 4). If water is dense with particles and cells, filter as much volume as possible and records the filtered volume.Cut each membrane in half with sterile surgical scissors and wrap it with aluminum foil. Proceed to DNA extraction with the half-cut membrane and store the other half in −20 °C as a back-up sample. Filtration of the water sample must be completed within 12 h of sampling; however, the filtered membranes can be stored at −20 °C for 4 weeks if DNA extraction cannot be performed on the same day.

#### 2.2.4. DNA Extraction

Extract DNA from the filtered membranes in Section 2.2.3 with the steps below. We selectively use the Chelex buffer method based on Nagai et al. [23].

Prepare 5% Chelex buffer with DNA/RNA free water in a sterile container. Heat a hotplate to 97 °C. Prepare three autoclaved 2-mL microtubes per sampling point.Place a half-cut filtered membrane in a 2-mL autoclaved microtube. Using sterile surgical scissors, cut the membrane into small pieces within the tube.Add 250 μL of 5% Chelex into the tube containing the membrane pieces and homogenize for 2 min for 1μm membrane and 3 min for 0.2 μm Sterivex membrane, respectively, with a Pellet Pestle™ Cordless Motor to break bacterial and phytoplankton cells (Figure 4).Add 250 μL of 5% Chelex into each tube to bring up the final volume of 500 μL. Heat membranes in tubes on a hotplate for 20 min at 97 °C. Transfer liquid to a new tube by pipetting. Label the tube and store it at −20 °C.

### 2.3. Analysis

This section describes the methods for metabarcoding, pigment, and nutrient analyses.

#### 2.3.1. Metabarcoding Analysis

The method was modified from the “16S Metagenomic Sequencing Library Preparation” (Illumina, Inc., San Diego, CA, USA) for 140–170 samples per batch run (Figure 5). Precautions: Clean pipettes and polymerase chain reaction (PCR) cabinet with 70% ethanol followed by UV light exposure for 30 min. Eight-channel pipettors, factory-sterilized filtered tips, and sterile tubes must be used. PCR reactions and primers should always be diluted from stock on each library preparation to avoid primer contamination.

First PCR reactions:Thaw the DNA samples from Section 2.2.4 at room temperature and use supernatant only for analysis.Prepare a master mix in a 1.5-mL tube containing the followings per a 25-μL reaction sample: 2.5 μL of 1 μM 16S-341F (16S) or SSU-F1289 (18S) primer, 2.5 μL of 1 μM 16S-805R (16S) or SSU-R1772 (18S) primer (Table 1), 12.5 μL of 2× Terra PCR Direct Buffer, 0.5 μL of Terra PCR Direct Polymerase Mix (1.25 U/μL), 5 μL of PCR grade water.Distribute 22.5 μL of the master mix into 8-tube strips and mix with 2.5 μL of DNA templates. The negative control is PCR grade water. Cap the strip immediately.Perform PCR on the 8-tube strips: Initial denaturation of 95 °C for 3 min, 35 cycles of 95 °C for 30 s followed by 55 °C for 30 s and 72 °C for 30 s, final elongation of 72 °C for 5 min. Store the PCR product at −20 °C until the next step is to proceed.

First PCR product confirmation and clean-up:Prepare a 100-mL of 2% *w/v* agarose-TBE gel containing 10 μL of GelRed^®^ Nucleic Acid Gel Stain. Mix 1.5 μL of 1× DNA loading dye with 4 μL of first PCR product, load the volume into the gel, run electrophoresis at 100 v for 30 min. Ensure the target bands at 500–600 bp, the primer-dimer bands at 80 bp, and no bands except primer-dimer in the negative control.Clean the first PCR product with Pronex^®^ Size-Selective Purification kit per the manufacturer’s manual. Transfer 20 μL of the final product to a new 96-well plate and seal it with a microseal film. Store at −20 °C until the next step is to proceed

Second PCR reactions and library verification:

This procedure is to amplify the cleaned first PCR amplicons with different primer combinations.

Dilute all index primers (Appendix Table A2) to 1 µM and align, as shown in Figure 6.In a new sterile 96-well plate, add the followings: 12.5 µL of 2× KAPA HiFi HotStart ReadyMix, 2.5 µL of each index primer (1 µM), 7.5 µL of purified PCR product DNA. Mix gently and cover with a microseal film.Perform PCR on the 96-well plate: Initial denaturation of 95 °C for 3 min, 8 cycles of 95 °C for 30 s followed by 55 °C for 30 s and 72 °C for 30 s, final elongation of 72 °C for 5 min. This “library” should be kept at −20 °C until the next step is to proceed.Perform library verification by the fragment analyzer, TapeStation 4000 series with 35–1000 bp reagents and D1000 sample buffer. The expected size of the final library is 613 bp for both 16S and 18S rRNA genes. Equilibrate D1000 Sample Buffer and ScreenTape at room temp for 30 min, and vortex and spin down. Mix 2 µL of D1000 Sample Buffer and 3 µL of the sample in new 8-tube strips, and verify the sample size by TapeStation 4000 series.

Second PCR clean-up, quantification, normalization:Clean the libraries by the method stated in the section of first PCR.Measure the DNA concentration in each second PCR product using Qubit4^TM^ coupled with Qubit^TM^ HS dsDNA kit. Accounting for the final library’s average size as 613 bp for both 16S and 18S rRNA genes, calculate DNA concentration in nM in each sample by:
(1)$$\frac{\mathrm{concentration}\;\mathrm{in}\;\mathrm{ng}/\text{µ}L}{660\frac{g}{\mathrm{mol}}\;\times \;630}\;\times \;10^{6}=\mathrm{concentration}\;\mathrm{in}\;\mathrm{nM}$$Dilute the libraries with sterile TE buffer (pH 8.0) to 4 nM in a new 0.2 mL 96-well plate. Store the plate at −20 °C until the next step is to proceed

Library denaturation:Thaw out Illumina MiSeq^TM^ reagent cartridge at 4 °C. Aliquot 5 µL of diluted library and mix all in a 1.5-mL tube as a pool on ice. Measure the DNA concentration of the pooled library. If higher than 4 nM (~1.6 ng/µL), it requires adjustment.Set a heat block at 96 °C. Place HT1 Hybridization buffer on ice. Dilute molecular grade NaOH from 1N to 0.2 N in a new tube. Dilute PhiX^TM^ from 10 nM stock to 4 nM in a new tube with TE Buffer pH 8.5.Mix 16 µL of the pooled libraries with 4 µL of PhiX^TM^. Mix 10 µL with 10 µL of 0.2 N NaOH in a new tube. Vortex this tube for 5 s, spin down briefly and incubate at room temperature for 5 min. Add 980 µL of HT1 Hybridization buffer to the tube and mix by inversion. Mix 260 µL of this solution with 390 µL of HT1 Hybridization buffer in a new tube and incubate it at 96 °C for 2 min followed by immediate transfer to ice for 2 min. The DNA concentration at this point is 8 pM.

Illumina MiSeq^TM^ sequencing: Set up the sample-sheet with each corresponding index i5 and i7 adapters on Illumina Experiment manager software. Clean the flowcell from the Illumina MiSeq^TM^ v3 kit with sterile molecular grade PCR water. Do not pour water directly on the capillary dots of the flowcell as it will get damaged. Gently wipe with non-fibrous paper. Remove MiSeq^TM^ v3 cartridge from 4 °C refrigerator. Load all the content (650 µL) of the samples into the MiSeq^TM^ cartridge. Start the program.

Metabarcoding data processing: Samples split into individual per-sample fastq files (demultiplexed samples) are processed using Dada2 [27]. Primer sequences are removed from amplicon reads using trimLeft option. The following steps (estimation of error rates, inference of sample sequences with the single-nucleotide resolution, merging paired reads, and taxonomic assignment) are conducted following the protocols of Dada2. Taxonomic assignment is conducted based on SSU Ref tree of SILVA release 132 [28].

#### 2.3.2. Pigment Analysis

The method for pigment analysis was modified from Sanz et al. [29]. The data interpretation follows Mackey et al. [30]. Pigment standards: Thirteen most basic pigments in CHEMTAX [30] are selected to study:Order pigment standards listed in Table 2 from DHI (Agern Allé 5, DK-2970 Hørsholm, Denmark) and obtain a Certificate of Analysis to record each standard’s actual concentration.Prepare two stock standard solutions, A and B. Standard solution A is prepared by mixing equal volumes of 19′But, Neo, Allo, Diato, and Zea. Standard solution B is prepared by mixing equal volumes of Perid, Fuco, 19′Hex, Prasino, Viola, Lut, Chl-*a*, and Chl-*b*.From the stock standard solutions, prepare two 7-point calibration curves A and B, ranging between 4 and 180 ng/mL by serial dilutions with 90% acetone.Prepare a standard mix by mixing equal volumes of stock standard solutions A and B. The pre-mixed pigment standard is used to verify the detection of each peak without a matrix effect. The standard solutions can be stored at −20 °C at least for three months.

##### 2.3.2.1. Pigment Extraction

Prepare samples as follows under subdued light. Soak the filtered GF/F membranes from Section 2.2.2 in 3 mL of 90% acetone in glass tubes for 15 min. Cut the membranes in acetone into pieces using sterile surgical scissors and sonicate for 5 min. Filter the product solution through a 13-mm diameter 0.2 μm PTFE membrane to remove cell debris. Transfer the filtrate into HPLC vials.

##### 2.3.2.2. HPLC-PDA Analysis

Photosynthetic pigments are quantified using a high resolution liquid chromatography system (HPLC) from Shimadzu (Kyoto, Japan) equipped with a Sil-10AF auto-sampler, a quaternary LC-10AT pump, DGU-14A degasser, CBM-20A System Controller, and SPD-M20A photodiode-array detector (PDA, 300–700 nm). Mobile phase A is a mixture of methanol and 225 mM ammonium acetate at 82:18 (*v/v*). Mobile phase B is ethanol. The HPLC column used is an ACE-1110-1546, (Advanced Chromatography Technologies, Aberdeen, Scotland). The monitoring wavelength for each pigment is listed in Table 3. The flow is 1 mL/min, column temperature is 40 °C, the sample temperature is ambient, and the injection volume is 15 μL. The gradient used is shown in Table 3. A re-equilibration time of 4 min should be set between injections.

##### 2.3.2.3. Data Processing

The representative chromatograms of a blank and standard mix are shown in Figure 7. Integrate peaks at the target wavelength. For example, use 446 nm to integrate a peak of 19′ but for both standard and samples. Record area under the curve (AUC) of standard and samples. Find Y-intercept and slope from the standard curve of each pigment. Calculate the amount of pigment detected per sample from the information:(2)$$pigment\;conc.\;=\frac{\mathrm{AUC}-\mathrm{Yintercept}}{\mathrm{slope}}\times \frac{\mathrm{vol}.\;\mathrm{HPLC}\;\mathrm{sample}\;\left(3\;\mathrm{mL}\right)}{\mathrm{vol}.\;\mathrm{sample}\;\mathrm{filtered}\;\left(1000\;\mathrm{mL}\right)}$$

#### 2.3.3. Nutrient Analysis

This section describes the dissolved nutrient analysis procedure to determine nitrite, nitrate, phosphate, and silicate in given seawater samples. The device to be used is an AQ400—discrete analyzer for environmental testing (SEAL Analytical, Inc., Mequon, WI, USA). The method was adopted from USEPA procedure 40 CFR part 136 and was modified for the AQ 400 discrete nutrient analyzer. Nitrite is determined by forming a reddish-purple azo dye produced at pH 2.0 to 2.5 by coupling diazotized sulfanilamide with *N*-(1-naphthyl)-ethylenediamine dihydrochloride (NEDD) with absorbance measured at 520 nm. Nitrate is determined by reducing nitrate to nitrite by buffer addition that reacts with sulfanilamide giving off a diazonium compound. This further reacts with *N-*(1-naphthyl) ethylenediamine dihydrochloride, forming a purple-red solution (520 nm). Phosphate is determined by reaction with acidic molybdate in the presence of antimony forming an antimony phospho-molybdate complex, which is further reduced by ascorbic acid and gives off a blue color measured at 880 nm. Silicate is determined under acidic conditions where silica combines with ammonium molybdate to form a yellow molybdous-silicic acid complex, which is reduced by 4-amino-3-hydroxy-1 naphthalene sulfonic acid to form a silico-molybdenum blue complex that is measured at 660 nm.

Prepare nitrite, nitrate, phosphate, and silicate standards per SEAL analytical test methods in Table 4. Construct standard curves with AQ400 following the manufacturer’s manual. Ensure each linear regression is above 0.9985, and the limit of detection (LOD) and limit of quantification (LOQ) meet the specification in Table 4. Thaw out the samples collected in Section 1 under subdued light and test the samples by AQ400 to obtain each dissolved nutrient value per the manufacturer’s manual.

### 2.4. Temperature and Precipitation

#### 2.4.1. Weather Station Data by Instituto de Fomento Pesquero (IFOP)

The information by IFOP is available from six metrological stations with data of temperature, precipitation, wind direction and intensity, atmospheric pressure, humidity, radiation, and PAR light in the following links:Comau fjord (https://www.hobolink.com/p/baa69a935234dc9d71febde8a42aa5a7),Apiao island (https://www.hobolink.com/p/7beb4f80e2fb5ee895848597e4b85cf7),Melinka (https://www.hobolink.com/p/ff1e12361933ece30d5de7902bf096ac),Cucao (https://www.hobolink.com/p/d84a8a7264813c76a812139d5ec42c0d),Putemún (https://www.hobolink.com/p/1fad3e888c904184f51fddcfe4202a71),Reloncaví fjord (https://www.hobolink.com/p/bd63a8b18b7e2990724e16ace75e3d41).

#### 2.4.2. Weather Station Data by Chilean Government

The metrological data are also available from open-access website provided by Dirección General de Aeronáutica Civil Dirección Meteorológica de Chile—Servicios Climáticos:


https://climatologia.meteochile.gob.cl/application/index/productos/RE7003


Download atmospheric temperature and precipitation vales of the area nearest to each sampling station.

### 2.5. Pilot Study

Following these protocols, the seawater sample was collected from Metri in Puerto Montt, Los Lagos region in southern Chile (−41°59′67″, −72°70′56″) on 24 April 2019, and assayed accordingly. The pigment assay was performed on the sample collected from the same station on 26 March 2019. Additionally, the presence of *Alexandrium* sp. and *Pseudochattonella* sp. in water samples from this station between January and May 2019 was monitored by microscope and 18S-rRNA gene analysis. The accession numbers of the fastq files obtained in this study are BioProject PRJNA668794, SRA codes SRR12814374 for 18S-rRNA and SRR12814375 for 16S-rRNA.

## 3. Results and Discussion

### 3.1. HAB Monitoring Methodologies

HAB monitoring procedures consist of many methodologies that include physical, chemical, and biological measurements of water samples and are currently available either separately in prior reports or reference quotations (Appendix Table A3). We introduced herein the step-wise procedures of HAB monitoring for the first time and compiled them in one standardized protocol to reduce errors and maintain data quality. Each methodology in the protocol set was carefully selected and optimized for specific reasons considering cost, materials, instrument availability, geographic accessibility, ease of use, and efficiency in productivity: The fourteen stations in Chile to be monitored for the prospective 4-year project have been selected based on their historical HAB records and accessibility. The microscopic observation uses the samples without filtration and fixation to identify *Pseudochattonella*, one of the notable HAB species in Chile with a small diameter that can be easily filtered out or ruptured with fixatives. The DNA extraction uses the Chelex-buffer method as Nagai et al. (2012) previously compared four different DNA extraction methods using *A. tamarense* and *A. catenella* and concluded that the 5% Chelex method was superior to others in efficiency, processing time, repeatability of PCR, and successful eukaryotic metabarcoding analysis [23,24,31,32]. The pigment analysis uses the thirteen most basic pigments because, of more than 70 pigments available, these are required by CHEMTAX, an algorithm program to estimate class abundances from pigment markers using pigment ratio [30]. Furthermore, the cost and time are significant considerations in pigment analysis as the standards are costly and must be imported for use in Chile due to only one qualified manufacturer that resides in Denmark. Thus, the pigments to be tested must be limited and carefully selected. For the nutrient analysis, caution must be taken in every step as a slight property change can affect the result. For instance, sterile 2-mL microtubes over larger containers are used to store water samples to avoid oxidation and ease of sample thawing and analysis processes. Following each step in the protocols for metabarcoding analysis helps to reduce human errors as it requires a long and complicated step using various reagents. The careful selection and optimization of the detailed procedures will undoubtedly improve the HAB monitoring project’s data quality.

### 3.2. Pilot Study: Metabarcoding Analyses

A pilot sampling was conducted at Metri in Puerto Montt on 26 March and 24 April 2019, and the water samples were processed and tested accordingly to demonstrate feasibility and productivity. The microscopic analysis of the water from 24 April 2019, quantified 682 phytoplankton cells/mL with the dominant species of *Lepidodinium chlorophorum*, a non-toxin producing dinoflagellate (Figure 8). There were also *Cheatoceros* sp., *Skeletonema* sp., *Eucampia zodiacus*, and *Leptocylndrus danicus* with >1% of the total cell count based on microscopic observations. The 18S rRNA gene metabarcoding analysis confirmed most of these species in the water sample but with many more phytoplankton genes such as *Gonyalax* sp. and *Thalassiosira* sp. (Figure 9). As compared to the microscopic analysis, the 18S rRNA gene metabarcoding analysis demonstrated the extensively broader capability of providing taxonomic information. However, due to its qualitative application, the metabarcoding analysis still needs to be coupled with microscopic analysis. One should also keep in mind that metabarcoding technology is costly and requires labor-intensive sample workups that could increase errors even with trained analysts [20]. Our metabarcoding analysis method can currently determine algal taxonomy to genus levels based on the genetic sequences. We are currently upgrading the method so that the same information can be utilized to obtain species-level information. This improvement will impact the understanding of responsible species for the HAB and damages to fishery activities.

Similarly, the 16S rRNA gene metabarcoding analysis was performed on the same water, and Figure 10 summarizes a list of bacterial genera detected as free-living bacteria in the water sample. How some of those bacteria may be potentially related to the present phytoplankton is currently unknown, but the predominant genus *Algiphilus* was previously reported as a dinoflagellate-associating phycosphere [33]. Exhaustive acquisition of phytoplankton 18S- and bacterial 16S-rRNA sequence information during the 4-year project period and data-processing using improved analysis method is expected to give significant insights into the algal-bacterial interaction that have a strong correlation with HAB dynamics.

*Pseudochattonella verruculosa* and *A. catenella* in Metri were monitored from January to May of 2019 by microscope and 18S rRNA gene metabarcoding analyses. While the microscopic analysis did not detect those species from Metri during the period, the 18S rRNA gene metabarcoding analysis identified the genus *Alexandrium* and *Pseudochattonella* periodically during the period with <1% relative abundance, although the species cannot be confirmed with our current method at this point (Figure 11). Metri, although historically rarely affected by HABs, was severely damaged by the “Godzilla-Red tide event” in the austral summer of 2016 that occurred in two stages, *P. verruculosa* bloom followed by *A. catenella* bloom, together resulted in great losses for the salmon industries and shellfish farms in the area [7,9,11]. With the method to be modified, we hope to track those two species in Metri by 18S rRNA gene metabarcoding analysis and to help with early warning detection.

### 3.3. Pilot Study: Physical and Chemical Analyses

Most of the physicochemical analysis results obtained from the pilot study were comparable to the prior reports suggesting that the protocols adopted here yield reliable data (Table 5). The salinity of the water from Metri (24 April 2019) was 30.2, corresponding to the lower end of the values reported by Sievers and Silva, who described that the salinities in this area are influenced by the freshwater sources contributed by rivers, glaciers, and coastal runoff [34,35]. Our measured water temperature was 12.0 °C on 24 April 2019, which coincides with April’s average water temperature in Puerto Montt recorded by the Weather Atlas. Considering that the atmospheric temperature of Puerto Montt on 24 April 2019, was 7.7 °C, this water temperature was relatively high. Aiken [34] explained that the warm water in this region corresponds to the flow from the Petrohué and Riñihue Rivers as the freshwaters tend to be associated with higher temperatures. Our measured DO at Metri on 24 April 2019, was 10.48 mg/L (117.7%), which was similar to the value reported by Aiken (Figure 7 in [36]), approximately 110% in Puerto Montt in early April of 2010. In general terms, the surface layer in southern Chile is reportedly well-oxygenated [37]. Our measured chl-*a* in the water was 15.05 μg/L, and the water transparency on the day was 3.5 m. These values were in line with Aiken [36], who reported a chl-*a* of 3 μg/L in Puerto Montt water in April 2010, during the non-bloom period (Figure 7 in [36]). It should be noted that chl-*a* values vary broadly based on the water condition, and it can be between a μg to mg per liter scale in southern Chile [38,39].

Our nutrient assay from the water at Metri on 24 April 2019, measured 0.04 μM nitrite, <LOD nitrate, 0.77 μM phosphate, and 17.13 μM silicate. Those values were similar to the report presented by Silva [37]. Interestingly, Silva [37] stated that the Seno Reloncaví, where Metri is located, receives significant influence from nearby rivers and glaciers, resulting in high silicate concentration in the surface water (20–100 μM) since freshwater is rich in silicate. Also, nutrient concentration in this region can easily fluctuate depending on many parameters such as freshwater contribution, turbulent mixing, biological production, respiration, and organic matter decomposition [36,37].

The pigment assay results for the water of Metri on 26 March 2019, were compared to the report by Wright et al. [40], who analyzed the waters from the transect between Antarctica and Australia in March 1987 using CHEMTAX for the first time (Table 5). All of our measured pigment values either fall in or are slightly higher than the ranges they reported. Some of our values are expected to be higher because Metri is on the coast where phytoplankton biomass is generally higher, while their samples are from the open-ocean. However, interpretation of pigment analysis data is not straightforward because many pigments such as Fuco, Chl-*b*, Zea, 19′Hex, and 19′But are present in several classes of phytoplankton while only a few markers are class-specific [40]. The CHEMTAX program came into place to solve this issue by using its algorithm. With the use of CHEMTAX, we foresee that at least a 4-years monitoring program will be needed to find a correlation between pigment by class (such as diatoms, dinoflagellates, haptophytes, prasinophytes, chlorophytes, and cryptophytes) along with other parameters.

## 4. Conclusions

We highlight the need to monitor algae and the associated microbiota interactions to understand bloom development and regressions over spatiotemporal scales. For this reason, a stepwise workflow of harmful algae monitoring in Chile is introduced to collect physical, chemical, and biological property data with emphasis on metabarcoding data to establish a sustainable HAB forecasting model. Having a set of standardized protocols is fundamental for collecting and comparing reliable HAB monitoring data from various sampling stations over a long study period. The present standardized protocols demonstrated feasibility and plausibility for sampling and analysis for the HAB monitoring. The water samples will continue to be collected from those selected stations in Chile for the duration using these protocols. Various statistical analyses are planned on the data to determine correlations in the measured parameters as related to the local HAB events. We believe such information will be useful to answer unresolved HAB questions as well as strategizing to mitigate damages on local aquaculture and fishery businesses caused by upcoming HABs. Lastly, we hope that sharing this step-wise standardized protocol with the public will be beneficial not only for our project but also for other researchers who are intensively studying HABs.

## Figures and Tables

**Figure 1 ijerph-17-07642-f001:**
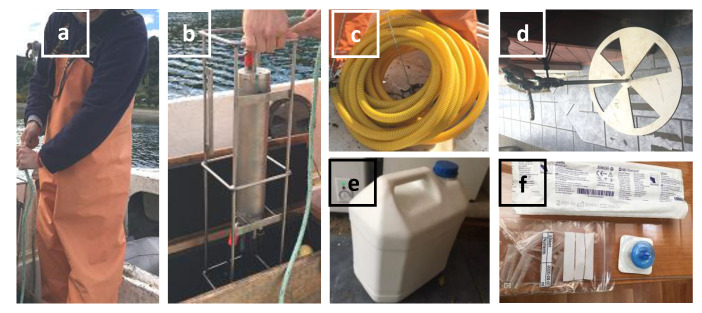
Materials and devices to carry to sampling sites: (**a**) fisherman’s waders and water-proof shoes, (**b**) CTD, (**c**) a 10-cm diameter with 10-m length siphoning hose with twisting lock on top and bottom, (**d**) Secchi disc, (**e**) a pre-cleaned 10-L plastic container, (**f**) sterile 50-mL syringe, 0.22 μm pore size filter, 5 × 2-mL sterile tubes, labeled bag and tubes with date, location, and sample ID.

**Figure 2 ijerph-17-07642-f002:**
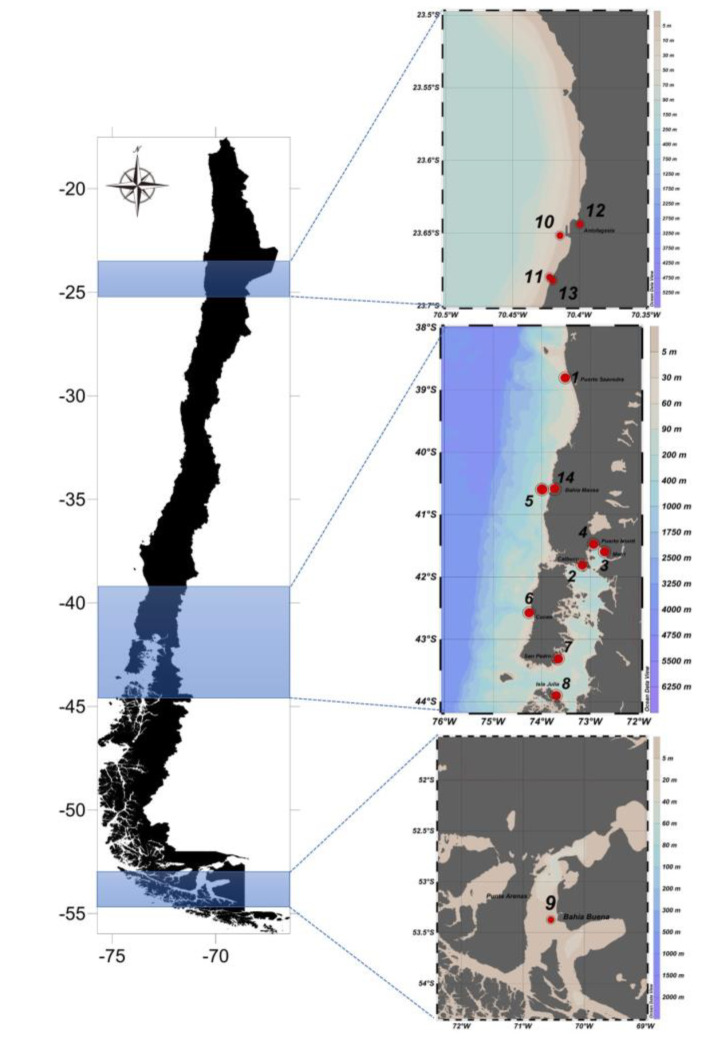
Sampling stations for prospective HAB monitoring program in Chile: at (1) Puerto Saavedra in Araucanía (−38.804; −73.516), (2) Paso Quenu in Calbuco (−41.807; −73.1672), (3) Metri in Puerto Montt (−41.597; −72.7056), (4) Club de Yates in Puerto Montt (−41.475; −72.934), (5) Bahía Mansa open ocean (−40.581; −73.737), (6) Cucao in Chiloé Island (−42.574; −74.259), (7) Isla San Pedro in Chiloé Island (−43.313; −73.662), (8) Isla Julia in Melinka (−43.901; −73.704), (9) Bahía Buena in Punta Arenas (−53.617; −70.914), (10) Puerto in Antofagasta (−23.652; −70.415), (11) Capilla in Antofagasta (−23.680; −70.423), (12) Puerto Costa in Antofagasta (−23.644; −70.399), (13) Capilla Costa in Antofagasta (−23.683; −70.420), (14) Bahía Mansa coast (−40.581; −73.737).

**Figure 3 ijerph-17-07642-f003:**
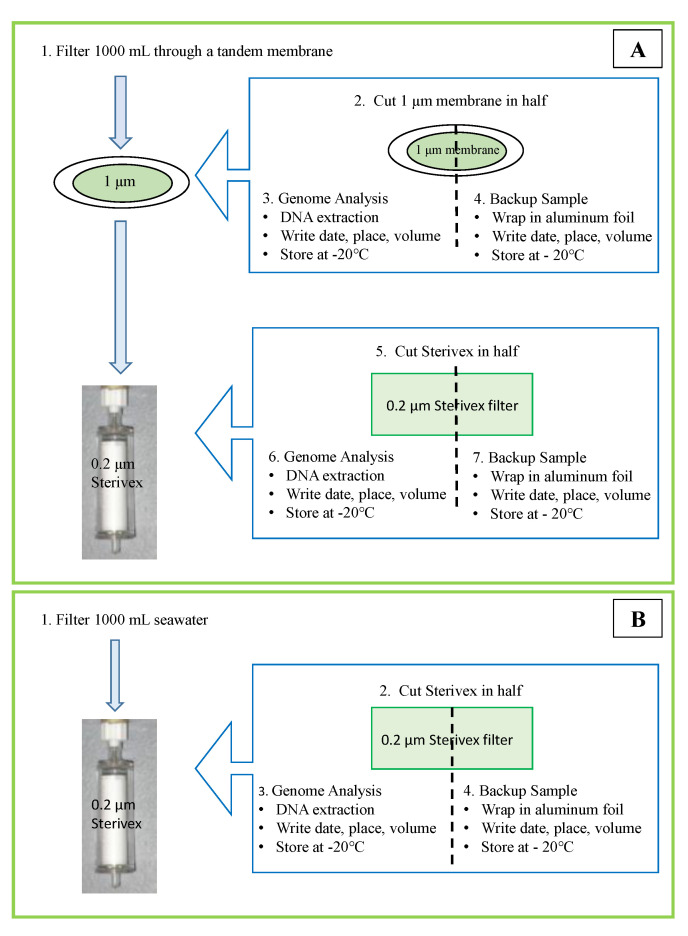
Sample preparation workflow for metabarcoding analysis: (**A**) 16S-rRNA: Water sample is filtered through a 1-μm pore-size membrane followed through a 0.2 μm pore-size membrane to capture attached and free-living bacteria, respectively. (**B**) 18S-rRNA: The water sample is filtered through only 0.2 μm pore-size membrane to capture phytoplankton. Each membrane is cut in half, one to be used for analysis and the other for back-up.

**Figure 4 ijerph-17-07642-f004:**
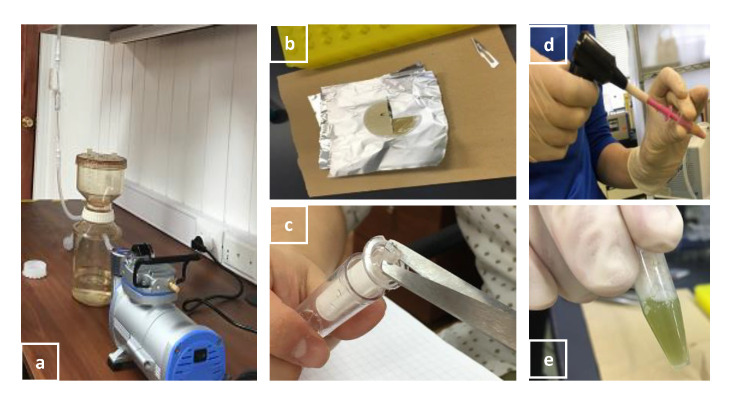
DNA extraction procedure: (**a**) filtration set up, (**b**) filtered one μm pore-sized membrane cut in half, (**c**) filtered 0.2 μm pore-sized Sterivex membrane, (**d**) homogenizing membrane in a tube with a Pellet Pestle™ Cordless Motor, and (**e**) DNA extraction product by 5% Chelex method.

**Figure 5 ijerph-17-07642-f005:**
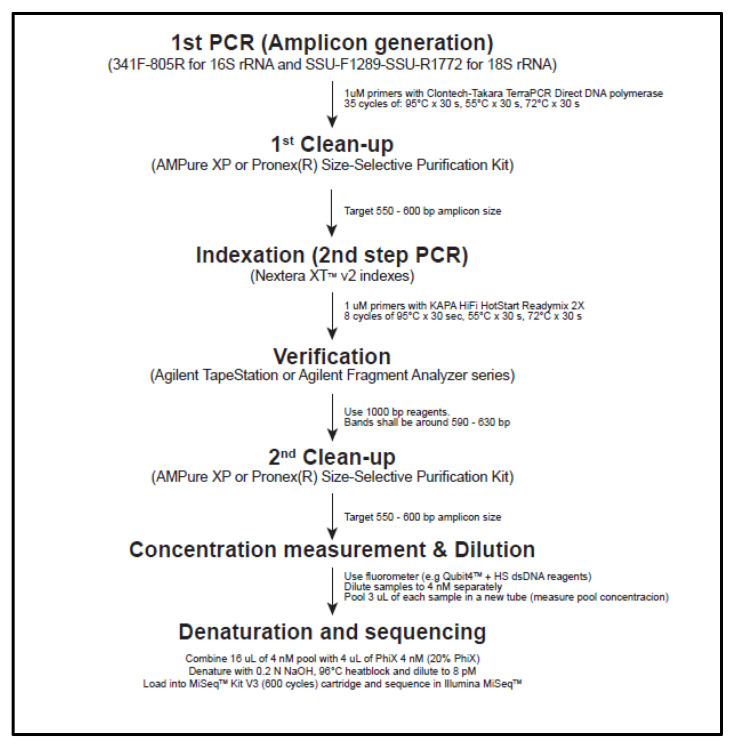
Metabarcoding analysis procedure: Extracted DNA is processed for the first and second PCR followed by sequencing and analysis.

**Figure 6 ijerph-17-07642-f006:**
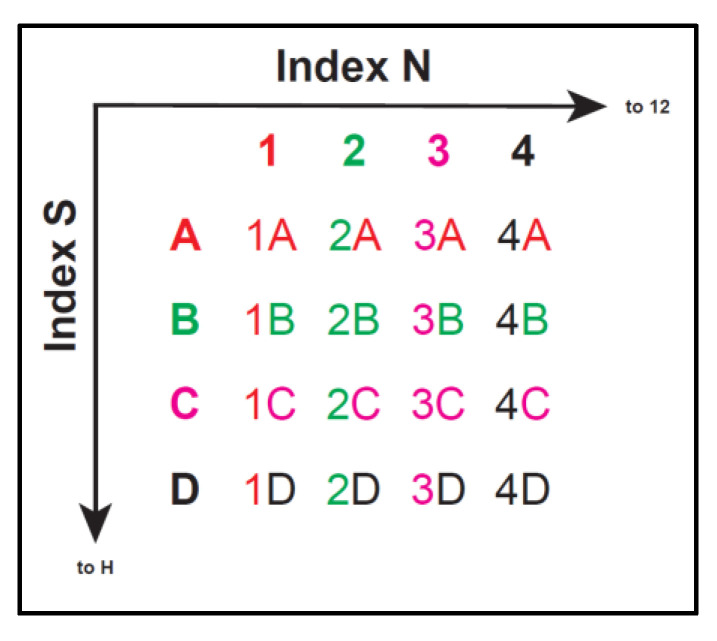
Exemplified sample alignment for MiSeq sequencing.

**Figure 7 ijerph-17-07642-f007:**
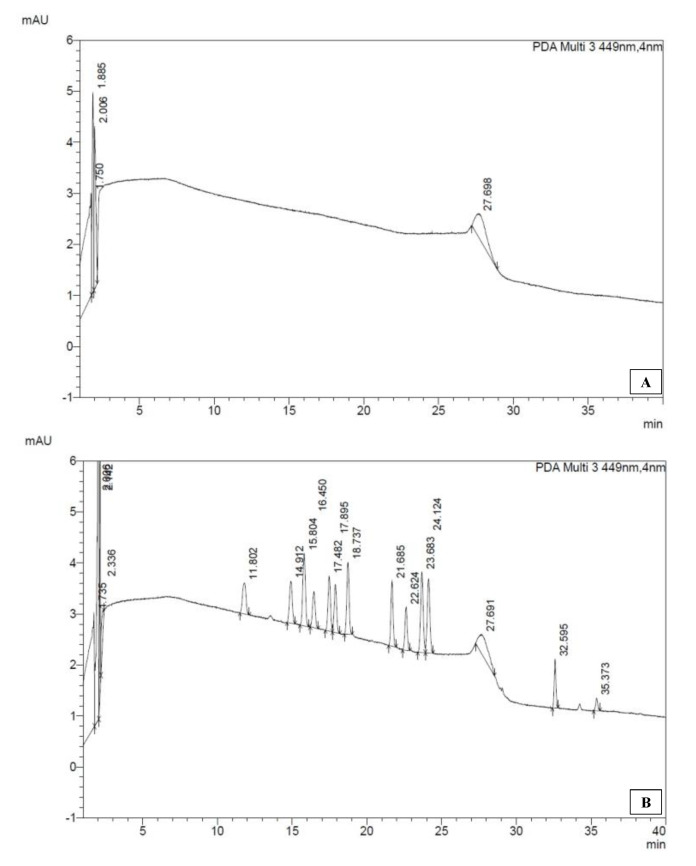
Representative chromatogram of pigments obtained by HPLC-PDA: Peaks detected by the wavelength of 449 nm from (**A**) water, and (**B**) standard mix. This wavelength was used to see all the peaks in a single chromatogram, but each pigment has maximum absorption at a different wavelength. Thus, each pigments quantification is carried out at the wavelength of maximum absorption (Table 2).

**Figure 8 ijerph-17-07642-f008:**
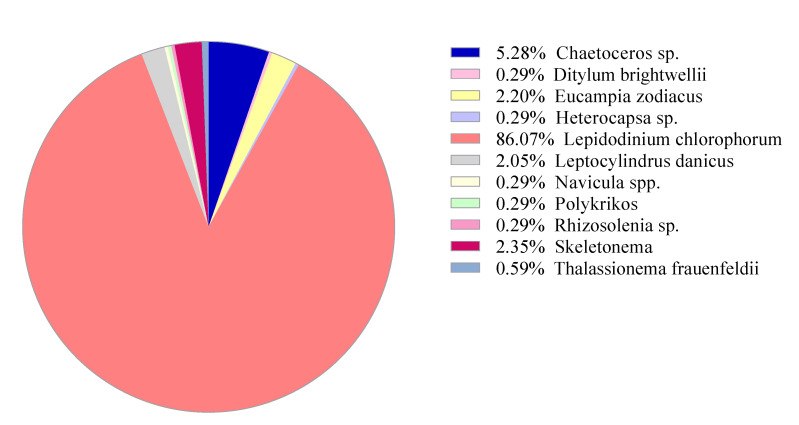
Microscopic phytoplankton identification and quantification by microscope: The water sample was collected from Metri, Chile (−41°5967′, −72°7056′) on 24 April 2019, from 0–10 m depth. A total of 682 phytoplankton cells were detected. *Lepidodinium chlorophorum* was the dominant species.

**Figure 9 ijerph-17-07642-f009:**
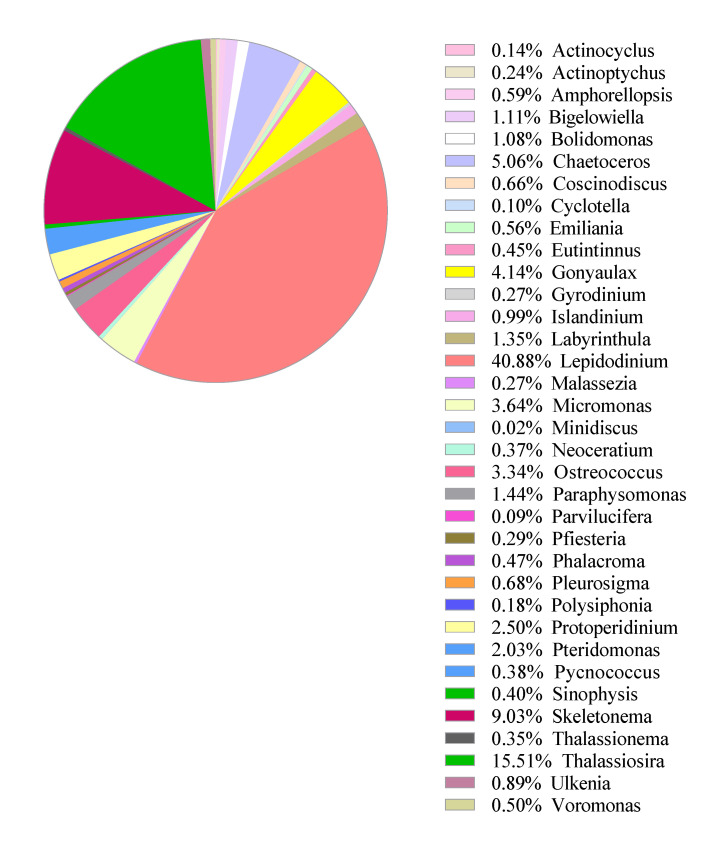
Phytoplankton relative abundance by 18S-rRNA gene metabarcoding analysis. The water sample was collected from Metri, Chile (−41°5967′, −72°7056′) on 24 April 2019, from 0–10 m depth. A total of 39,330 read pairs were obtained.

**Figure 10 ijerph-17-07642-f010:**
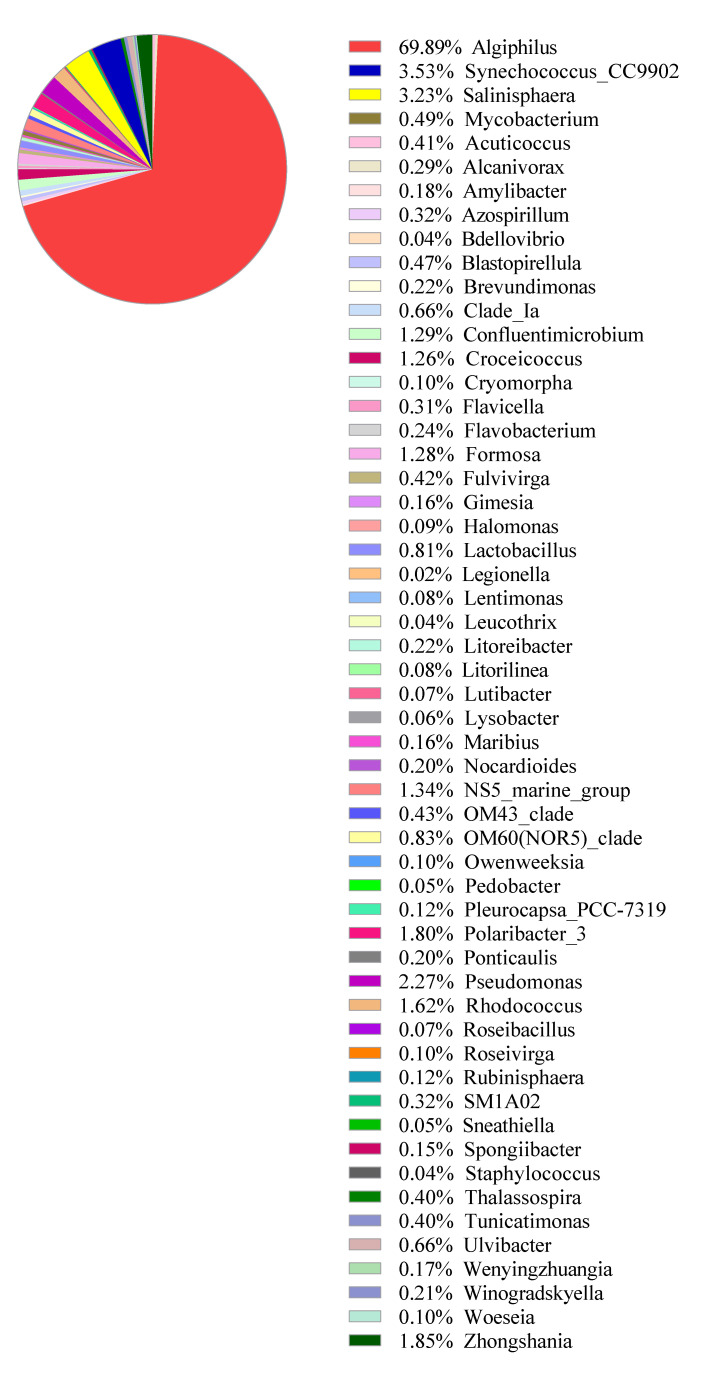
Free-living bacteria relative abundance by 16S-rRNA metabarcoding analysis. The water sample was collected from Metri, Chile (−41°5967′, −72°7056′) on 24 April 2019, from 0–10 m depth and filtered through a 0.2 μm pore-sized membrane. A total of 28,635 read pairs were obtained.

**Figure 11 ijerph-17-07642-f011:**
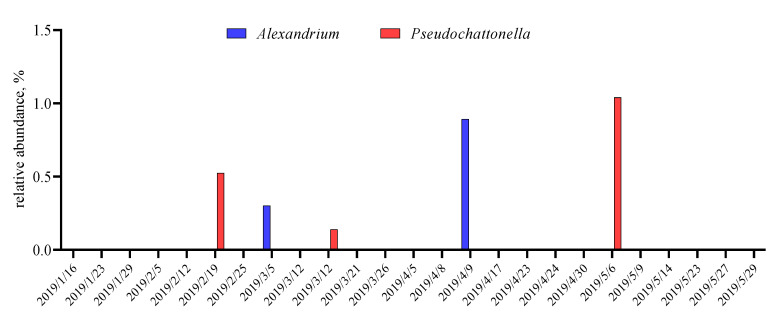
Gene detection of the genus of *Alexandrium* and *Pseudochattonella* by 18S-rRNA metabarcoding analysis. The water sample was regularly collected from Metri, Chile (−41°5967′, −72°7056′) in the fall of 2019. The genes of *Alexandrium* and *Pseudochattonella* were detected at some time-points, although below 1% of relative abundance.

**Table 1 ijerph-17-07642-t001:** PCR primers selected for metagenomic library preparation for first PCR reactions.

Primer Name	Overhang Adapter (5′ → 3′)	Region of Interest Specific Sequence (5′ → 3′)	Ref.
SSU-F1289	ACACTCTTTCCCTACACGACGCTCTTCCGATCT	TGGAGYGATHTGTCTGGTTDATTCCG	[24,25]
SSU-R1772	GTGACTGGAGTTCAGACGTGTGCTCTTCCGATCT	TCACCTACGGAWACCTTGTTACG	
16S-341F	ACACTCTTTCCCTACACGACGCTCTTCCGATCT	CCTACGGGNGGCWGCAG	[26]
16S-805R	GTGACTGGAGTTCAGACGTGTGCTCTTCCGATCT	GACTACHVGGGTATCTAATCC	

**Table 2 ijerph-17-07642-t002:** Pigment standards.Thirteen most basic pigment standards are used for analysis as required by Mackey et al. [30]. Each vial comes with 2.5 mL at approximately 1 mg/mL from DHI. Certificate of Analysis must be obtained from DHI for the actual concentration of each standard. Retention time is expected to shift slightly depending on the condition.

Standard	Abbreviation	Target Wavelength	Retention Time
19′-Butanoyloxyfucoxanthin	19′But	446 nm	14.9
19′-Hexanoyloxyfucoxanthin	19′Hex	447 nm	17.5
Alloxanthin	Allo	453 nm	21.7
Chlorophyll-*a*	Chl-*a*	665 nm	35.4
Chlorophyll-*b*	Chl-*b*	466 nm	32.6
Diatoxanthin	Diato	449 nm	22.6
Fucoxanthin	Fuco	449 nm	15.8
Lutein	Lut	445 nm	24.1
9′-*cis*-Neoxanthin	Neo	439 nm	16.5
Peridinin	Perid	472 nm	11.8
Prasinoxanthin	Prasino	454 nm	17.9
Violaxanthin	Viola	443 nm	18.7
Zeaxanthin	Zea	450 nm	23.7

**Table 3 ijerph-17-07642-t003:** Gradient set up for pigment analysis by HPLC-PDA. Mobile phase A is methanol: 225 mM ammonium acetate = 82:18 (*v/v*). Mobile phase B is ethanol.

Time (min)	Mobile Phase A (%)	Mobile Phase B (%)
0	96	4
0.2	96	4
16	62	38
22	62	38
22.1	28	72
30	20	80
36	10	90
36.1	96	4

**Table 4 ijerph-17-07642-t004:** Nutrient analysis standard and test methods. Prepare standards to construct a seven or 8-point calibration curve per the test methods. Ensure each standard curve passes the acceptance criteria of r, limit of detection, and quantification limit. The standard information is listed in Appendix Table A1.

	Nitrite	Nitrite-Nitrate	Phosphate	Silicate
Standard	NNaO_3_	NNaO_3_	KH_2_PO_4_	Na_2_O_3_Si·9H_2_O
Standard curve	7-point 0.001–0.2 mg/L	8-point 0.01–2.0 mg/L	7-point 0.003–0.3 mg/L	8-point 0.10–10 mg/L
*r* acceptance	>0.9985	>0.9985	>0.9985	>0.9985
LOQ	0.001 mg/L	0.01 mg/L	0.003 mg/L	0.10 mg/L
LOD	0.002 mg/L	0.003 mg/L	0.0006 mg/L	0.018 mg/L
USEPA number	600/R 93/100, 1993.	600/R 93/100, 1993.	600/R 93/100, 1993.	600/4-79-020, 1983.
SEAL AQ2 number	EPA-137-A	EPA-127-A	EPA-155-A	EPA-122-A

**Table 5 ijerph-17-07642-t005:** Seawater properties at Metri, Chile. On 24 April 2019, the water sample was collected for physical and dissolved nutrient analyses, and on 26 April 2019, for pigment analysis. The values for CTD data at Metri are an average from 0–10 m depth. The atmospheric data were obtained from El Tepual airport in Puerto Montt, the nearest available metrological station to Metri. The reference values for pigment are taken from the plot scales used in Wright et al. [40].

Assay	Parameter	Value at Metri	Reference
atmospheric	temperature	7.7 °C	NA
	precipitation	None	NA
Secchi	depth transparency	3.5 m	NA
CTD	water temperature	12.0 °C	12 °C (weather atlas)
	salinity	30.2	31–33 PSU [34]
	dissolve oxygen	10.48 mg/L (117.7%)	110% [36]
	Chlorophyll-*a*	15.05 μg/L	3 μg/L [36]
nutrient	NO_2_	0.04 μM	NA
	NO_3_	<LOD	0–8 μM [37]
	PO_4_	0.77 μM	0–8 μM [37]
	Si(OH)_2_	17.13 μM	20–100 μM [37]
pigment	19′But	0 μg/L	0–0.08 μg/L [40]
	19′Hex	0.113 μg/L	0–0.15 μg/L
	Allo	0 μg/L	0–0.01 μg/L
	Chl-*a*	2.073 μg/L	0–0.6 μg/L
	Chl-*b*	0.341 μg/L	0–0.15 μg/L
	Diato	0 μg/L	NA
	Fuco	0.568 μg/L	0–0.15 μg/L
	Lut	0μg/L	0–0.01 μg/L
	Neo	0 μg/L	NA
	Perid	0.027 μg/L	0–0.08 μg/L
	Prasino	0.025μg/L	0–0.01 μg/L
	Viola	0.014 μg/L	NA
	Zea	0 μg/L	0–0.01 μg/L

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
