# Peer review of "Protocols for Monitoring Harmful Algal Blooms for Sustainable Aquaculture and Coastal Fisheries in Chile"

_ijerph, 2020, doi:10.3390/ijerph17207642_

Round 1

Reviewer 1 Report

Harmful algal blooms are routine ecological disasters for many coastal areas, as they are harmful to both human health and ecosystem sustainability. However, how to routinely monitor these events is still not fully solved. Scientists are managing to establish some standard procedures so that results from different locations and under different environmental conditions could be equivalently compared.

Yarimizu and other authors made this step forward by setting and testing their proposed procedure in field samples in Chilean coastal areas. The whole procedure is thorough that covers almost all aspects of essential parameters. And it is plausible these authors emphasize the important role of associated microbial consortium that was neglected in traditional studies.

If anything these authors could make to revise this manuscript, I would suggest them: (1) rewriting abstract for a balanced standard form with background, method, result and implication, (2) comparing the proposed procedure with other available monitoring protocols. Particularly for the second point, it is vital to do this evaluation as to propose a standardized procedure is claimed as the primary goal.

Author Response

Thank you for reviewing our manuscript and providing your suggestions. We rewrote abstract in the order of background, method, results, and implication.  To respond to your advice (2), we added Appendix C listing the current major HAB monitoring programs. None of these programs disclose their HAB monitoring protocols, however. Therefore, we could not make a Table to show a comparison between our protocols and theirs.  Each organization and research group has its methodology, and the only way to follow is to read publications. But, publications usually state brief descriptions about methods of HAB monitoring. We hope this explanation answers your advice (2). Please also see the revised manuscript (uploaded). Thank you again for your patience. 

Reviewer 2 Report

This is a very interesting protocol that, I am sure, will be of great use in the close future. Nevertheless, this manuscript needs to be improved.

English must be revised, preferably by a native speaker.

Line 34: change for use different combinations.

L42: change collect for obtain.

L52, 55, 56: there are “the” that should be erased.

L57: were

L64: analysis

L65: ongoing

L76: is responsible for certificating the products. Or product certification.

L76: the authors translated all governmental names to English, with few exceptions. Be consistent, translate all or none.

L82: Bell and Mitchell (year).

L95: basis.

L104: the information obtained…

L109: Information on materials, reagents and instruments is listed in Appendix A.

L145: specify why 1 L of sample.

L149: how can you store immediately at -20 °C? If you are in a vessel, sampling for some hours?

L150: The use of nitrile gloves should be stated in line 109.

L161: how are these samples taken?

L166: mL

L167: how can we transfer 12 mL if instructions before said 10 mL?

L168: But there are other trademarks that are as good as Falcon, right? Why not just state a 15 mL plastic centrifuge tube?

L170: do you refer to a Sedgewick Rafter?

L176: 1 L. Specification of the filter is missing.

L178: and record the filtered volume.

L180: it is not possible to filter 1 L of seawater directly through a 0.2 µm membrane.

L182: membranes should be cut with sterile scissors?

L185: how many weeks are “few weeks”? Be as specific as possible.

L187: 1 µm

L189: all phytoplankton cells are larger than bacteria, unless you refer to planktonic bacteria

L195: erase “one”.

L196: clean or sterile?

L268: erase “of it”

L269: 5 s.

Table 2: Capitalize first letters. You report a retention time but did not mention the type of detector used.

L306-307: Unintelligible.

L312: It is a very common mistake to report the separation technique (HPLC) as the analytical method. No. Chromatography will ONLY separate the components of a mixture. The real analysis is performed by the DETECTOR. So you should always report precisely which detector you are using: HPLC-UV-vis.

L319: same comment.

L323: punctuation is missing.

Figure 7: You state: “… chromatogram of pigment assay by HPLC”. NO. You did not analyze by HPLC. You never mentioned the detector. Also, those baselines are very poor, or the peaks are too short. Use a better chromatogram.

L358: State what IFOP means.

L380: Accession numbers usually should be uploaded previous to submitting the paper.

L394: add the year.

L404: What do you mean by “Eppendorf tubes”? What volume? Sterile? Maybe you mean 1.5 mL sterile microtubes? Or maybe 15 mL centrifuge tubes? Trademark is not important, but other features are.

L413: specify that the dominant species is a dinoflagellate. That is important even if it was not a toxin-producer.

L423: change all references to “pipeline” to “method”. It is not a pipe.

L429: finding mostly a bacterial genus reported as dinoflagellate-associated bacteria is not surprising since the dominant species was a dinoflagellate.

Figure 8: Microscopic identification…

L436: was the dominant species.

L443: do not start a sentence with P. State the genus.

L446: erase “of”

L453, Alexandrium should be in italics.

L458: pilot’s study or pilot study

L460: salinity does not need units.

L463, 464, 465: what do you mean by “on the day”?

L470: water transparency.

Table 5. Capitalize on first letters. What does that * mean?

L485: erase “the”.

L486: can easily fluctuate.

L511: to answer.

Appendix A: this is a very useful table, but it would be more useful if the authors list first all the required equipment, and its average cost. Also, the average cost of all these materials, and the number of people needed to perform these assays (technicians).

Author Response

Thank you for patiently reviewing our manuscript and providing us detailed suggestions. We appreciate your comment on valuing our protocols. It meant a lot to us. We are responding to your comments in the Table in Word file because it is easier to follow this way. The responses to your suggestions are also highlighted in blue in the revised manuscript. We hope those corrections meet your expectation. Thank you againf for your patience. 
